# Development and Evaluation of a Self-Assessment Tool for Family Caregivers: A Step Toward Empowering Family Members

**DOI:** 10.3390/nursrep15110385

**Published:** 2025-10-29

**Authors:** Laura Schwedler, Jan P. Ehlers, Thomas Ostermann, Gregor Hohenberg

**Affiliations:** 1Stabsstelle für Digitalisierung und Wissensmanagement, Hochschule Hamm-Lippstadt, 59063 Hamm, Germany; gregor.hohenberg@hshl.de; 2Fakultät für Gesundheit, Universität Witten-Herdecke, 58455 Witten, Germany; jan.ehlers@uni-wh.de (J.P.E.); thomas.ostermann@uni-wh.de (T.O.)

**Keywords:** caregivers, self-assessment, stress, psychological, nursing, quality of life

## Abstract

**Background/Objectives:** Family members who provide care play a central but often underestimated role in the healthcare system and are frequently exposed to considerable physical, emotional, and social stress. To better understand and support their needs, a structured self-assessment tool (SSA-PA) was developed. This development addresses the current lack of practical, validated instruments that enable caregivers to systematically reflect on their own stress levels and resources, which is becoming increasingly important in view of the growing demand for care and the risk of caregiver burnout. This tool aims to promote self-reflection, identify individual stresses and resources, and enable more targeted support for family caregivers. **Methods**: The development process (September–December 2024) followed a multi-phase design that integrated theoretical foundations from nursing, health, and psychology, in particular Orem’s theory of self-care deficit, Lazarus and Folkman’s stress and coping model, and Engel’s biopsychosocial model. Four core dimensions were defined: (1) health and self-care, (2) burden and stress, (3) support and resources, and (4) satisfaction and quality of life. The final tool comprises 37 items (mostly 5-point Likert scales), supplemented by multiple-choice and open-ended questions. Content validity was ensured through expert review and testing with nine family caregivers. Internal consistency was assessed using Cronbach’s alpha (α = 0.998), indicating very high reliability, although possible item redundancies were identified. The evaluation took place in January 2025 with 33 family caregivers to assess user-friendliness, relevance, and perceived usefulness. **Results**: The majority of participants rated the tool as user-friendly and clearly structured. Around 80% reported a high level of comprehensibility, and over half stated that the tool provided new insights into their own health and care burden. Qualitative feedback highlighted the value of the tool for self-reflection and motivation to seek external support. Suggestions for improvement included shorter item formulations, improved visual feedback (e.g., progress bars or charts), and expanded question areas on financial burdens and digital support options. **Conclusions**: The SSA-PA is a theoretically grounded and user-centered tool for assessing and reflecting on the situation of family caregivers. It not only enables systematic self-assessments but also promotes awareness and proactive coping strategies. Future research should focus on conducting factor analyses to further validate the construct, testing the tool in larger samples, and exploring its integration into structured care consultations to improve the quality of home care.

## 1. Introduction

The care of relatives by family members and close friends is an essential but often underestimated pillar of the healthcare system. In Germany, family caregivers provide the majority of home-based care services for people in need of assistance, thereby making a substantial contribution to relieving the professional care sector. These informal caregivers make a significant contribution to social welfare by taking on tasks that can be extremely demanding, both physically and mentally [1]. The continuous care they provide to their relatives in need of care not only affects their well-being but also has a profound impact on the quality of life of the caregivers themselves [2]. Research shows that caregivers are disproportionately affected by exhaustion, depression, social isolation, and physical complaints. Despite the fundamental role they play within the social and health care network, they often lack the necessary support and recognition that would do justice to their efforts [3]. In particular, systematic support structures that help them reflect on their own burdens and resources are still insufficiently developed [4].

Against this backdrop, the development of structured tools for self-assessment of individual care situations is becoming increasingly important. The SSA-PA was developed by the authors specifically with the aim of providing a theoretically sound, user-centered instrument with established validity and reliability that fills a gap in the tools available for comprehensively supporting self-reflection among caregivers and targeted interventions. Such tools enable family caregivers to systematically record their workload, available resources, and support needs, thereby developing a better awareness of their own situation. The physical, mental, and psychological health of family caregivers is crucial, as the stresses of care-giving can often lead to health problems. Studies show that family caregivers are at in-creased risk of chronic diseases and psychological distress such as depression [5]. Access to appropriate support structures is essential for family caregivers to avoid burnout. Resources such as social support, financial assistance, and access to care services are crucial for making care sustainable [6]. Caring for relatives is often associated with high physical and emotional stress. Stress can lead to burnout and other health problems, which impair caregivers’ ability to perform their tasks [7]. The quality of life and satisfaction of family caregivers influence not only their willingness to provide care, but also the quality of care they provide. A high level of life satisfaction can help increase resilience to stress and strain, thereby making the caregiving experience more positive [8]. For healthcare professionals, these tools can provide a valuable basis for counseling sessions, training courses, and individual support measures.

To date, various international and national questionnaires have been developed to assess the burden on family caregivers—for example, the Zarit Burden Interview (ZBI) [9,10,11], the Caregiver Strain Index (CSI) [12,13,14], and the CarerQol instrument [15,16,17]. These instruments are valuable tools for assessing overload or quality of life, but they often only capture certain aspects of the caregiving situation. In particular, the dimensions of self-care, social support, and subjective stress perception are only taken into account to a limited extent. Against this background, there is a need to develop an tool that provides a more comprehensive and practical picture of the individual situation of family caregivers.

This article describes the development and initial evaluation of a self-assessment tool for family caregivers (SSA-PA), which enables a structured and comprehensive assessment of stress, resources, and support needs. The aim of the study was to systematically present the development of the SSA-PA, examine its content validity, comprehensibility, and user-friendliness, and obtain initial indications of its reliability.

## 2. Materials and Methods

### 2.1. Study Design

This study follows an exploratory mixed-methods design that combines both quantitative and qualitative elements to comprehensively capture the development, theoretical foundation, and initial evaluation of an online self-assessment tool for family caregivers (Self-Assessment Tool for Family Caregivers, SSA-PA). The self-assessment tool was developed using a systematic and scientifically sound approach with the aim of creating a tool that is both theoretically sound and practice-oriented. This tool is designed to capture the diverse stresses and needs of family caregivers.

The development process followed a clear chronological sequence consisting of three consecutive phases:

(1) Theoretical conceptualization and a literature review;

(2) Item formulation and validation;

(3) Evaluation by family caregivers.

In addition, experts from nursing science, psychology, and health services research, as well as family caregivers themselves, were involved in the development process to ensure the practicality and relevance of the tool.

### 2.2. Development of the Tool

Phase 1: Theoretical conceptualization and literature review

The development of the SSA-PA was based on sound theoretical foundations from the fields of nursing science, health science, and psychology. The central theoretical reference points were Dorothea Orem’s nursing model, Lazarus and Folkman’s stress and coping model, and Engel’s biopsychosocial model.

Orem’s concept of self-care emphasizes the ability and responsibility of individuals to actively contribute to their own health and well-being. Orem’s nursing concept pro-vides a basis for identifying support needs by emphasizing the importance of self-care and self-help [18].In addition, psychological theories on stress management were applied. Among other things, the stress management model developed by Lazarus and Folkman provides a sound basis by emphasizing cognitive assessment and coping strategies in dealing with stress [19].Engel’s biopsychosocial model offers a comprehensive perspective by incorporating physical health factors, psychological states, and social contexts [20,21].

To provide empirical support for these theoretical models, a structured literature search was conducted in the PubMed, CINAHL, PsycINFO, and CareLit databases. Search terms (German/English) used included:

“family caregivers” OR “informal caregivers” AND “burden” OR “self-assessment” OR “questionnaire” OR “self-care”.

Search parameters and selection process:Time frame: The search period generally covered publications from 2015 to 2024 in order to take into account current scientific findings and recent instrument developments. Older sources were only included if they provided fundamental theoretical or empirical contributions to the stress, self-care, or coping of family caregivers and were therefore indispensable to the theoretical foundation of the SSA-PA.Languages: Only studies published in English or German were included.Types of publications: Both quantitative and qualitative studies, systematic reviews, meta-analyses, and instrument development studies were considered.Inclusion criteria: Studies focusing on informal (non-professional) caregivers of adults or older persons receiving home-based care. Research examining burden, stress, coping, self-care, quality of life, or support needs among family caregivers. Publications describing theoretical frameworks or measurement instruments. Studies providing psychometric validation, conceptual models, or dimensions relevant to caregiver well-being and self-care.Exclusion criteria: Studies focusing exclusively on professional caregivers, institutional care settings, or pediatric caregiving. Publications without empirical or theoretical relevance to the constructs of burden, self-care, or coping (e.g., policy papers, editorials, commentaries). Conference abstracts, letters, and opinion pieces without full methodological transparency.

Of a total of 137 studies identified, 29 papers were included in the conceptualization. From these, four central dimensions were derived that represent the essential areas of life of family caregivers:The dimension “health status and well-being” includes questions on physical and mental health in order to assess general well-being and health-related stress. For example, Schulz and Sherwood (2018) emphasize the significant impact of caregiving on both physical and mental health, highlighting the chronic stress experienced by caregivers [22]. Similarly, Savage and Bailey (2004) discuss the mental health effects of family caregiving, underscoring the importance of assessing these aspects [23]. Other studies also emphasize the considerable physical and psychological strain associated with caregiving tasks, as well as the resulting chronic exhaustion and emotional stress experienced by family caregivers [24,25,26,27,28].The dimension “burden and stress” highlights the everyday challenges and stress factors that family caregivers face. McIlvennan et al. (2021) explore the stress and coping mechanisms among family caregivers, particularly in high-stress situations such as caring for patients with a left ventricular assist device [29]. Additionally, Uğurlu et al. (2024) examine the relationship between compassion, stress, and coping strategies in caregivers of patients with heart failure, emphasizing the emotional burden and stress experienced by caregivers [30]. Other studies also emphasize that coping with stress-related burdens in particular plays a central role in the physical and mental health of family caregivers and thus contributes significantly to promoting resilience and prevention [31,32,33,34,35].The dimension “support needs and resources” assesses the need for external support and the use of available resources. Berry et al. (2017) propose a framework for supporting family caregivers of cancer patients, highlighting the importance of assessing caregivers’ needs, educating them, empowering them, and providing proactive assistance [36]. Shillam (2022) discusses legislative efforts to provide resources and support for family caregivers, emphasizing the need for a national caregiving strategy [37]. Further studies also confirm the importance of perceived support from formal and informal networks, as well as satisfaction with respite care and nursing services, for the well-being of family caregivers [28,38,39,40,41,42,43,44].The dimension “satisfaction and quality of life” aims to assess subjective quality of life and satisfaction. Perpiñá-Galvañ et al. (2019) examine the burden and health-related quality of life of caregivers of palliative care patients, identifying key factors that predict caregiver burden and its impact on health [45]. Gan et al. (2022) examine the quality of life of family caregivers of cancer patients in a developing country, highlighting the significant psychological stress and low quality of life experienced by caregivers [46]. Further studies also confirm the importance of life satisfaction, a sense of purpose, social participation, and emotional resilience as key factors for the well-being and long-term stability of family caregivers [47,48,49].

These dimensions reflect both the stress and resource perspectives and form the theoretical basis for the formulation of the question items.

Phase 2: Item formulation and validation

Based on the theoretical foundations and the identified dimensions, a structured questionnaire was developed. The items were formulated in a multi-stage, theory- and practice-based process that was carried out from September 2024 to December 2024. The aim was to develop a content-based, comprehensible, and practical self-assessment tool for family caregivers.

The item development was divided into several steps:Initial item generation:

Based on theoretical models (Orem, Lazarus & Folkman, and Engel) as well as relevant literature and existing validated instruments (e.g., the Zarit Burden Interview, CarerQoL, and the COPE Index), questions were first formulated that reflect the four central dimensions of the tool: health status and well-being, burden and stress, support needs and resources as well as satisfaction and quality of life.

2.Expert validation:

Three experts from the fields of nursing science, psychology, and health services research evaluated the items in terms of comprehensibility, relevance, and theoretical fit. The focus was particularly on ensuring content validity and theoretical coverage. The expert panel consisted of two males and one female, aged between their mid-20s and mid-50s.

3.Target group participation:

To ensure that the tool was practical and comprehensible in everyday use, nine family caregivers were involved in its development. In moderated focus groups, they evaluated the language, emotional appeal, and content suitability of the questions. This feedback enabled the wording and topics to be adapted to the actual experiences of family caregivers in a way that was appropriate for the target group.

4.Revision and fine-tuning:

Based on feedback from experts and the target group, duplicate questions were removed, unclear terms were simplified, and scale options were standardized. In addition, cognitive pre-tests were conducted to check comprehensibility and response behavior. Feedback was incorporated iteratively into the revision of the questions.

The items were predominantly assigned five-point Likert scales (“does not apply at all”–“applies completely”) in order to capture intensities and perceptions in a differentiated manner. In addition, multiple-choice questions (e.g., on the use of support services) and free text fields were used to allow for individual reflections and personal additions.

The final structure of the Self-Assessment for Family Caregivers (SSA-PA) comprises four content sections with a total of 37 items. This structure supports thematically coherent and user-friendly use, both online and in counseling situations.

Table 1 shows the structure of the SSA-PA for family caregivers. The table shows the individual dimensions of the tool, the corresponding questions, and the response formats used (Likert scales, multiple-choice, or open-ended questions). This makes the systematic classification of the survey content and response types within the questionnaire transparent and easy to understand.

Phase 3: Evaluation by family caregivers

An online survey of 33 family caregivers was conducted to test and evaluate the practicality of the SSA-PA. Participants assessed aspects such as user-friendliness, comprehensibility of the questions, and relevance of the content. They were also able to provide suggestions for improvement in free-text fields. The majority of respondents rated the comprehensibility and applicability of the tool positively (see Figure 1 and Figure 2).

The evaluation served not only as a practical test but also as preparation for a later psychometric validation with a larger sample. The evaluation of the self-assessment tool was conducted in January 2025.

The results of the evaluation are presented in detail in Section 3 (“Results”).

### 2.3. Data Analysis

The quantitative data were analyzed using IBM SPSS Statistics (version 30). Descriptive statistics (means, standard deviations, frequencies) were calculated to describe the sample and evaluate the item responses. The internal consistency of the instrument was determined using Cronbach’s alpha.

The overall instrument yielded a very high value of α = 0.998, indicating an exceptionally high degree of homogeneity among the items. However, such a high alpha value can also indicate redundancy among the items or overlap in content. Therefore, the items were reviewed in terms of content to identify possible overlaps. Some questions showed similarities in content (e.g., repeated wording on stress and exhaustion), indicating the possibility of item consolidation in future versions.

Cronbach’s alpha was calculated based on the closed questions (Likert and multiple-choice items) of the sample of 33 family caregivers. Open-ended questions were not included in the reliability analysis as their content was evaluated qualitatively. A separate calculation of Cronbach’s alpha for the individual dimensions was not performed due to the small sample size, but is planned for a follow-up study.

No factor analyses (exploratory or confirmatory) are available to date to verify validity, as this would require a larger sample size. However, content validity was ensured by the multi-stage development process, which included feedback from both experts and family caregivers.

Content validity index (CVI)

To further verify content validity, the content validity index (CVI) was calculated according to Lynn (1986) [50]. Three experts from the fields of nursing science, health psychology, and nursing education independently assessed the relevance of the individual items on a four-point scale (1 = not relevant to 4 = highly relevant). The item-level CVI (I-CVI) was calculated as the proportion of experts who rated each item with a 3 or 4 (see Table 2).

Overall Results:

S-CVI/Ave = 0.892

S-CVI/UA = 0.73

Total agreement (items with full agreement among all experts) = 27

These values indicate a high degree of content validity, as an S-CVI/Ave above 0.80 is considered acceptable for newly developed instruments. Only a few items (e.g., 2, 4, 11, 14, 20, 24) were identified as potentially ambiguous and will be revised in future versions to improve clarity.

Statistical analysis of quantitative data

Non-parametric tests were used for the items on the Likert scale in order to take account of the ordinal level of measurement. Correlation analyses were performed to investigate correlations between key dimensions such as stress, self-care, and need for support. In addition, group comparisons (e.g., by gender, relationship to the care recipient, and duration of care) were performed using appropriate nonparametric methods. The analyses revealed meaningful correlations between several core dimensions and showed that a longer duration of care was associated with higher perceived stress.

The multiple-choice questions were evaluated using descriptive statistics to determine the distribution of responses and identify prevailing trends. The results showed that, after using the instrument, the majority of participants became more aware of their own need for support and were motivated to seek external support.

Qualitative analysis

The validity of the open-ended questions was assessed using a qualitative content analysis according to Mayring (2015) [51]. The extent to which the responses reflected the theoretically intended dimensions and corresponded in content to the closed-ended questions was examined. The qualitative analysis showed a high degree of consistency between the main topics addressed in the open-ended responses and the four theoretical dimensions of the instrument (health status, stress, need for support, and quality of life). This suggests that the open-ended questions are suitable for capturing and deepening the intended constructs.

## 3. Results

The following results reflect the evaluation of the self-assessment tool developed for family caregivers. The evaluation was conducted to assess the user-friendliness, relevance, and acceptance of the tool and to gain valuable insights for its further development.

### 3.1. Participant Characteristics

The evaluation of the self-assessment tool was carried out in January 2025 with 33 family caregivers who had previously completed the tool in its entirety. The aim was to obtain as diverse a picture of the user group as possible.

The participants comprised different age groups, educational levels, and caregiving experiences (see Table 3). The average age of the respondents was just over 40. 78.8% of the respondents were female, 18.2% male, and 3% diverse.

With regard to educational attainment,

36.4% reported having completed vocational training;21.2% reported having a university degree;15.2% reported having a general university entrance qualification;12.1% reported having a secondary school diploma;9.1% reported having another educational qualification;and 6.1% reported having a secondary school diploma.

The participants’ caregiving experience varied significantly:33.3% had been caring for their relative for less than six months;30.3% for more than five years;21.2% for one to five years;and 15.2% for six to twelve months.

For most (48.5%) family caregivers, the time spent on caregiving was 10–20 h per week, but for 21.2% of respondents, it was more than 30 h per week.

This heterogeneous composition makes it possible to take into account feedback from family caregivers in different life and care situations. It thus forms a solid basis for evaluating the user-friendliness, relevance, and acceptance of the tool.

### 3.2. User-Friendliness and Clarity of the Tool

A key objective of the evaluation was to review the user-friendliness and linguistic comprehensibility of the self-assessment tool (SSA-PA) that had been developed. Respondents evaluated both the general handling and the comprehensibility and structure of the individual questions. The clearly formulated questions and intuitive navigation through the tool were designed to ensure that it could be easily used by family caregivers of all ages and educational levels.

The results show a high level of acceptance and user-friendliness of the tool.

Around 78% of participants described the use of the SSA-PA as “very easy” or “easy,” while 19% rated it as ‘neutral’ and only 3% as “difficult” (see Figure 1). This assessment is also reflected in the qualitative comments, which particularly highlighted the clear structure, the thematic division into four dimensions, and the visually clear design.

When assessing the clarity and comprehensibility of the wording, 78% of respondents stated that the questions were “very clear” or “clear” (see Figure 2). The everyday language used was particularly praised, as it enabled family caregivers to relate to the questions.

However, some participants suggested improvements regarding the length of the tool and expressed a desire for clearer visual navigation, for example, through progress bars or color highlighting of the four subject areas. This feedback was incorporated into the planned revision of the online version of the SSA-PA (see Table 4).

The overwhelmingly positive feedback confirms that the tool is easily accessible to family caregivers and can be used intuitively in both written and digital form. This user-friendliness is a crucial prerequisite for the acceptance and sustainable use of the SSA-PA.

### 3.3. Relevance of the Questions

A key objective of the evaluation was to assess the relevance of the individual questions in the SSA-PA in terms of the participants’ living and care situations. The assessments provide valuable insights into the extent to which the items formulated reflect the actual needs, burdens, and resources of family caregivers.

The results show that the majority of participants (around 54%) rated the questions as “very relevant” or “relevant” to their personal care situation (see Figure 3). Only a small proportion (around 12%) stated that individual questions were less relevant, while 33% were neutral on the subject. Questions relating to self-care, emotional stability, and coping with stress were particularly positively highlighted. These were perceived as directly relevant to practice and helped participants to reflect on their own care situation in a more differentiated way.

In addition, many participants stated that the tool had encouraged them to reflect on their own role as caregivers and to become more aware of previously neglected aspects of self-care and relief (see Figure 4 and Figure 5). These results demonstrate that the tool not only serves to record stress levels but also functions as a tool for reflection and learning.

Qualitative feedback shows that respondents particularly appreciated the complexity of the topics. Several participants emphasized that the questions “realistically reflect everyday life” and “encourage reflection”. At the same time, some expressed a desire for additional items that take greater account of specific care situations (e.g., dementia, Parkinson’s disease, or multiple illnesses) (see Table 5). This feedback will be used for future enhancements to the tool in order to develop disease-specific modules.

The results regarding the relevance of the items thus confirm that the theoretically derived dimensions (health and self-care, stress and strain, support and resources, satisfaction and quality of life) are also perceived as central and accurate from the user’s perspective. This speaks for the content validity of the tool and supports the chosen theoretical foundation based on the Orem model, the stress model according to Lazarus & Folkman, and the biopsychosocial approach according to Engel.

### 3.4. New Findings and Support

An important goal of the evaluation was to examine the extent to which the self-assessment tool functions not only as a survey tool, but also as a reflection and learning aid for family caregivers. The focus was on whether completing the SSA-PA helped participants gain new insights into their own health, stress, and self-care, as well as identify support needs.

The results show that half of the participants (around 51%) stated that answering the questions had given them new insights into their own caregiving situation (see Figure 6). Many said that the tool encouraged them to be more aware of their own well-being and personal stress levels. Questions about self-care and stress perception in particular were described as “eye-opening” because they focused on aspects of everyday caregiving that had previously received little attention.

The tool also had a significant impact in terms of identifying support and relief needs. As Figure 4 illustrates, over 60% of respondents reported that working with the tool motivated them to consider external relief options.

Figure 5 also shows that around 69% of participants became aware of support services through the tool and want to make use of them.

These results suggest that the SSA-PA has an activating function beyond mere self-assessment: it promotes awareness of one’s own situation and encourages people to consider their individual resources and limitations.

The tool can therefore be understood not only as a diagnostic tool, but also as a preventive and advisory support tool.

Qualitative feedback underscores this finding: several participants reported that working with the tool had made them “think consciously about their own overload for the first time” or “encouraged them to accept help”.

One participant stated: “I realized that I am part of the care process myself and that I can’t just look after others”. The feedback shown in Table 6 shows that the tool promotes self-reflection, awareness of one’s own situation, and health awareness among participants. Many reported that they had gained new perspectives on their stress, their role as family caregivers, and their well-being. At the same time, participants identified areas for improvement: suggestions included placing greater emphasis on self-care in the evaluations, integrating individual feedback after completion of the tool, providing more specific recommendations for promoting health and well-being, and adding practical tips or examples for implementing self-care. These suggestions provide valuable insights for the further development of the tool in order to support users’ ability to reflect and take action in an even more targeted manner.

These results also confirm the usefulness of the SSA-PA from a theoretical perspective. In line with Orem’s self-care model and Lazarus and Folkman’s stress and coping approach, the tool supports self-reflection as a prerequisite for adaptive coping strategies and health-promoting behavior. The feedback makes it clear that structured self-assessment is often the only way for family caregivers to recognize which resources and support needs are actually available or unused.

### 3.5. Completeness of the Tool

Another key aspect of the evaluation was to assess the completeness of the content of the self-assessment tool. The aim was to check whether the tool adequately covers the essential areas of the life and care situation of family caregivers and whether participants feel that topics relevant to their individual care situation are missing.

The results paint an overall positive picture. As Figure 7 shows, around 51% of participants rated the tool as “complete” or “very complete” in terms of the topics covered. Only a small group of around 3% felt that the tool was “not complete at all”.

Feedback from participants showed that the tool in its current form was perceived as comprehensive and holistic. Particular praise was given to the fact that the SSA-PA not only focuses on nursing activities, but also takes into account health, emotional, and social aspects of the care situation. One participant summed this up aptly:

“I found it very helpful that the questions were not only about physical strain, but also about satisfaction and what is good for me personally”.

This perception underlines the coherence of the four dimensions–health & self-care, strain & stress, support & resources, and satisfaction & quality of life–and confirms their relevance for the holistic self-reflection of family caregivers.

At the same time, some participants pointed out gaps in content and potential for improvement. These related in particular to financial burdens, legal issues, digital support services, and dealing with feelings of guilt in everyday caregiving—topics that were only marginally addressed in the first version of the tool. The desire for more specific reflection questions and a visually appealing presentation of results (e.g., diagrams, progress bars, or profile displays) was also expressed several times.

A systematic evaluation of the open feedback revealed thematically consistent suggestions for improvement, which are summarized in Table 7. They show that the SSA-PA already covers a broad spectrum of relevant areas of life, but could be expanded to include social-legal, digital, and spiritual-social dimensions in order to better reflect the diversity of individual care situations.

This feedback confirms that the tool is fundamentally perceived as complete, but can be further refined to accommodate new developments, particularly in the area of digital support.

From a methodological perspective, the results underscore that the content validity of the tool is given, but at the same time needs-based further developments are useful in order to better reflect the heterogeneity of family caregivers.

### 3.6. Recommendations and Suggestions for Improvement

A key objective of the evaluation was to obtain feedback on the practical applicability and potential areas for improvement of the self-assessment tool that had been developed. In particular, the focus was on whether family caregivers would recommend the tool to others and what adjustments would be useful to increase its comprehensibility, relevance, and user-friendliness.

As Figure 8 shows, 69% of participants stated that they would “definitely” or “probably” recommend the tool to other family caregivers. This high recommendation rate illustrates that the self-assessment tool (SSA-PA) in its current form enjoys a high level of acceptance and perceived usefulness. About 21% of respondents were undecided (“maybe”), while only a small minority of 9% said they would be unlikely to recommend the tool to others.

In the qualitative feedback, many participants said that they found the tool “very helpful for their own reflection”. Several family caregivers stated that working with the tool had made them think consciously about their own stress, self-care, and actual support needs for the first time. One participant said:

“I had never taken the time to think about my own situation in such detail before. The tool made me realize that I urgently need more breaks”.

In addition to this predominantly positive feedback, several specific suggestions for improvement were collected in the open question round, which provide valuable insights for future development. Frequently mentioned were:Shortening the processing time: Some respondents found the questionnaire to be relatively long, especially when they also had caregiving responsibilities. A modular structure or adaptive versions could remedy this.Visual presentation of results: The desire for clear feedback in the form of graphical representations, progress bars, or self-assessment profiles was expressed several times.Expansion of the subject areas: Suggestions related to additional items on financial burdens, legal support, and digital aids in caregiving.Individual feedback: Several participants wanted an automated evaluation with personalized recommendations, e.g., on relief services or counseling centers.

These results show that the SSA-PA is perceived by the target group as a relevant and practical tool, but at the same time there is potential for structured further development. In particular, the desire for interactive elements and personalized feedback underscores the need to make the tool digitally adaptive in the future and to integrate it more strongly into counseling situations in accordance with § 45 SGB XI.

Overall, the results show that the SSA-PA is viewed positively by family caregivers as a holistic, practical, and adaptable tool. The identified suggestions for improvement form a solid basis for further development into a digital, adaptive version that can promote self-reflection among family caregivers in an even more targeted manner in the future.

### 3.7. Summary of Results

Overall, the evaluation of the SSA-PA showed that the tool was highly accepted, understandable, and relevant. The majority of the 33 participants rated the user-friendliness as “very easy” or ‘easy’ and the clarity of the wording as “very clear” or “clear”. These results confirm that the tool is well suited to the target group and can be used without prior specialist knowledge.

Respondents also gave predominantly positive feedback regarding the relevance of the content. The questions were perceived as relevant to the individual care situation and encouraged reflection on health, stress, and available resources. The holistic approach, which takes physical, emotional, and social dimensions into account equally, was particularly appreciated.

More than half of the participants reported that the tool had given them new insights into their well-being and self-care. Many stated that they had thought more consciously about personal limits and support needs, which underlines the potential of the SSA-PA as a basis for reflection and discussion in care counseling.

Cronbach’s alpha of 0.998 indicates very high internal consistency and thus strong content coherence of the items. However, for future developments, a factor-analytical review would be useful to check for possible redundancies and strengthen construct validity.

Overall, the evaluation confirms that the SSA-PA is a reliable, practical, and user-oriented tool that supports family caregivers in systematically reflecting on their situation.

## 4. Discussion

The development and evaluation of the Self-Assessment Tool for Family Caregivers (SSA-PA) represents an important step toward strengthening a target group that has often been neglected in the past. Family caregivers take on central tasks in home care and are exposed to considerable physical, emotional, and social stress [39,52]. The SSA-PA offers, for the first time, a systematic and holistic way of recording and reflecting on the individual stress, available resources, and support needs of these individuals in a structured manner.

The tool covers four dimensions—health and self-care, stress and strain, support and resources, and satisfaction and quality of life—and thus reflects the essential aspects of everyday caregiving experience. Through a combination of Likert scales, multiple-choice questions, and free text fields, the tool enables a differentiated, both quantitative and qualitative assessment of the individual care situation. This methodological diversity helps to reflect the complexity of the care experience and to derive individual recommendations for action.

The evaluation confirmed the user-friendliness and comprehensibility of the tool. The majority of respondents found the questions to be clearly formulated, practical, and relevant to their personal care situation. Many stated that the tool encouraged them to become more aware of their own stress, self-care, and social support. This demonstrates the potential of the SSA-PA as a basis for reflection and discussion in care counseling in accordance with § 45 SGB XI. Other potential areas of application include:Individual use: Promotion of self-reflection, recognition of stress and support needs, basis for discussions with professionals [53].Health and social services: Systematic needs assessment for the development of tailored support services, e.g., training, psychological care, or resilience promotion [54,55].Policy and care: Aggregated data can reveal gaps in care, support financial and legal measures, and improve the legal framework [25,56].Research and teaching: Provision of quantitative and qualitative data on the stress and resources of family caregivers; integration into training programs for professionals [57].

Comparable approaches to assessing the burden and needs of family caregivers have been presented in previous research. Instruments such as the Zarit Burden Interview (ZBI) [9,10,11], the Caregiver Strain Index (CSI) [12,13,14], and the COPE Index [15,16,17] are among the best-known instruments for measuring the burden and support of family caregivers. However, these instruments focus primarily on the quantitative assessment of stress and do not adequately address aspects of self-care, health behavior, or subjective well-being. In contrast, the SSA-PA combines these dimensions, offering a broader, more holistic perspective on the caregiving experience.

Similar to the findings of Schulz and Martire (2004) [6] and Wilz and Pfeiffer (2019) [39], who emphasize that self-reflection and early recognition of personal stress can help prevent a deterioration in the health of caregivers, the present evaluation also shows that the SSA-PA promotes awareness and self-care behavior. Furthermore, studies by Pinquart and Sörensen (2003) [5] and Berry et al. (2017) [36] have shown that structured self-assessment interventions can strengthen coping strategies and increase the willingness to seek external support—effects that are also reflected in the qualitative feedback from the participants in this study.

Accordingly, the SSA-PA builds on existing instruments and expands them by integrating reflective and preventive elements that enable caregivers to not only assess their own situation, but also to actively interpret it. This supports the growing consensus in current nursing research that promoting self-awareness and self-efficacy is a central component of sustainable nursing support.

The calculated Cronbach’s alpha of 0.998 indicates exceptionally high internal consistency. Although this result speaks for the homogeneity of the items, it also points to possible duplications or overlaps in content. Further exploratory or confirmatory factor analysis is therefore necessary to verify construct validity and identify redundant items. Furthermore, no test–retest reliability has been collected to date, so it is not yet possible to make statements about the temporal stability of the results. The validity of the open-ended questions also requires qualitative analysis to assess the extent to which they adequately reflect the intended concepts.

The participants provided valuable feedback in the evaluation, which will contribute to the further development of the tool. In particular, they mentioned the desire for shorter items, clearer visualizations (e.g., progress bars, results diagrams), and additional topics such as financial burdens, digital support options, and feelings of guilt in everyday caregiving. These suggestions underscore the importance of continuously revising the tool to adapt it even more closely to the reality of life for family caregivers.

Despite the positive feedback, some limitations must be taken into account. The sample size of 33 participants is limited and restricts the statistical significance and generalizability of the results. In addition, the data was collected in an online survey, which does not rule out self-selection bias. External validation of the tool is still pending. In addition, practical problems may arise when using the SSA-PA. Some family caregivers may have difficulty completing the instrument independently due to cognitive or emotional limitations. Technical problems, time constraints, or limited reading and writing skills may also hinder its use. In addition, cultural and linguistic differences may require adjustments to ensure that the instrument is accessible and meaningful to different population groups. Future studies should therefore include larger and more heterogeneous samples to test the psychometric quality and applicability of the SSA-PA in different contexts.

The tool provides a solid basis for future research. Research could be conducted to examine the extent to which regular use of the SSA-PA improves the self-management, quality of life, and well-being of family caregivers in the long term. It should also be analyzed how the results of the tool can be integrated into counseling processes and used to derive individualized support measures. In addition, the implementation of the tool in care counseling could be examined in order to improve the quality of care and counseling practice. Of particular interest would also be validation in an international context and the digital further development of the tool, for example, as an interactive online tool or app, in order to further increase accessibility and acceptance.

In summary, the discussion shows that the SSA-PA is a promising, practice-oriented, and theoretically sound tool. It enables family caregivers to reflect on their own situation in a differentiated way and supports professionals in counseling to make individual stress profiles visible. Through the planned further validation steps and optimizations, the SSA-PA can make an important contribution to promoting self-care and relieving the burden on family caregivers in the future.

## 5. Conclusions

The self-assessment tool developed for family caregivers (SSA-PA) is a practical and scientifically sound tool that makes an important contribution to promoting self-reflection and self-care in home care. It enables family caregivers to systematically assess their individual care situation and thus identify stressors, resources, and support needs. The tool thus creates the basis for targeted relief measures and more individualized counseling.

The innovative added value of the SSA-PA lies in its holistic approach, which integrates four central dimensions: health and self-care, burden and stress, support and resources, and satisfaction and quality of life. This comprehensive view enables a differentiated analysis of the complex care situation and contributes to improving the quality of life of family caregivers. The evaluation showed that the tool was perceived as understandable, relevant, and helpful. It supports family caregivers in becoming more aware of their own needs and at the same time offers professionals a valuable basis for structuring counseling sessions in a structured and needs-oriented manner.

Despite the positive results, there are limitations, particularly with regard to the small sample size and the lack of external validation to date. These aspects limit the generalizability of the results and should be addressed in future studies. Nevertheless, the SSA-PA provides a solid basis for further research and development.

Future studies should focus on validating the tool with larger and more heterogeneous samples and examine how the SSA-PA influences the well-being and self-care of family caregivers in the long term. Similarly, integration into care counseling in accordance with Section 45 of SGB XI is a promising approach to improving counseling processes and sustainably increasing the quality of support for family caregivers. In addition, aggregated results could help to manage regional support structures in a more targeted manner and identify needs at an early stage.

Overall, the SSA-PA marks a significant step forward in the recording and support of family caregivers. It raises awareness of their own situation, opens up opportunities for action, and provides valuable impetus for research, practice, and policy. The tool thus makes a lasting contribution to the recognition and empowerment of those who perform the majority of care work in Germany.

## Figures and Tables

**Figure 1 nursrep-15-00385-f001:**
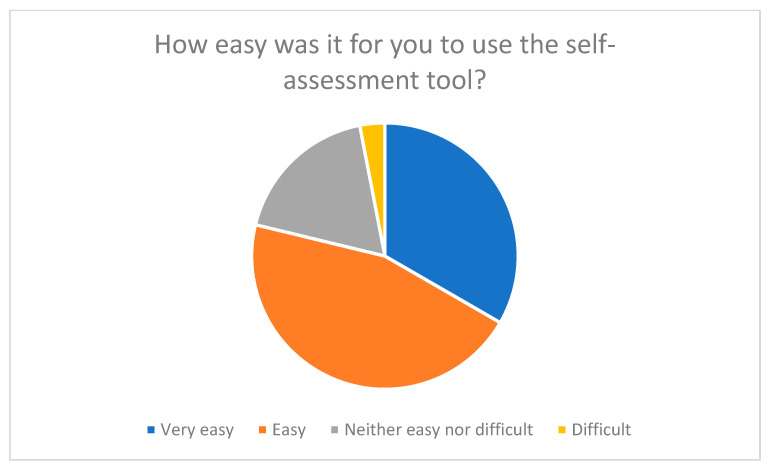
User-friendliness of the self-assessment tool from the perspective of respondents.

**Figure 2 nursrep-15-00385-f002:**
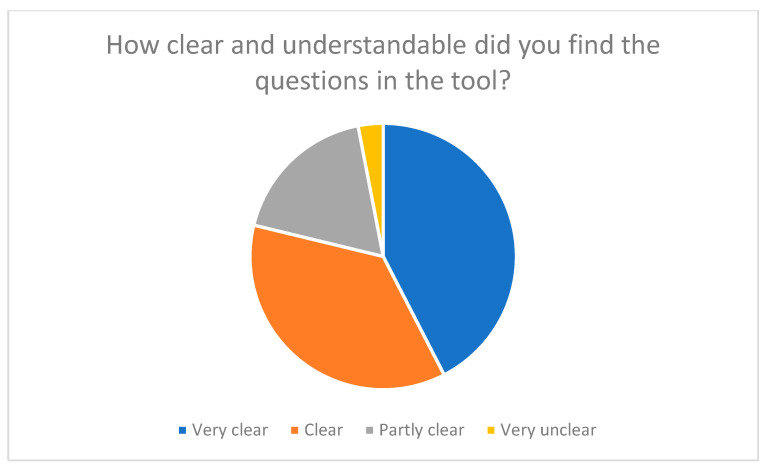
Clarity and comprehensibility of the questions in the tool.

**Figure 3 nursrep-15-00385-f003:**
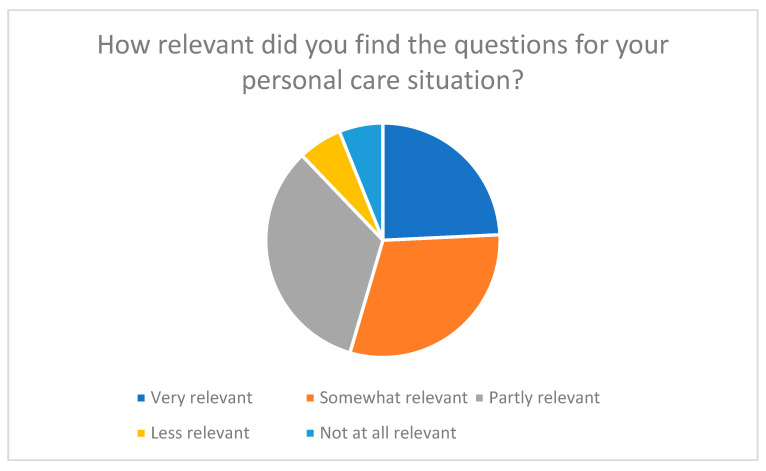
Relevance of questions in relation to personal care situation.

**Figure 4 nursrep-15-00385-f004:**
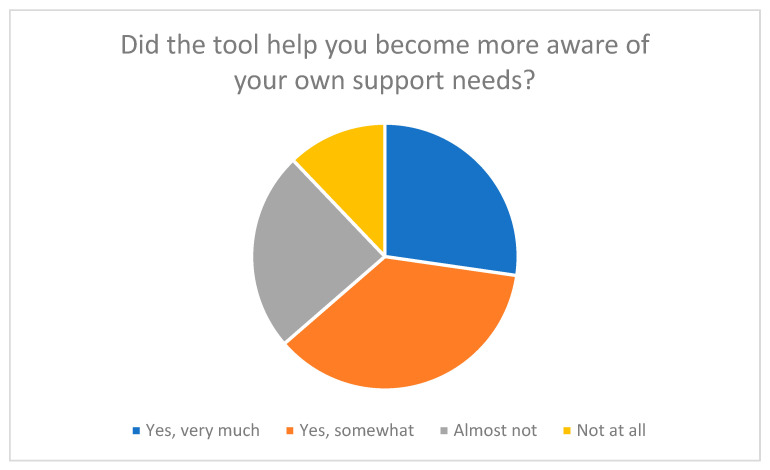
Raising awareness of one’s own support needs through the tool.

**Figure 5 nursrep-15-00385-f005:**
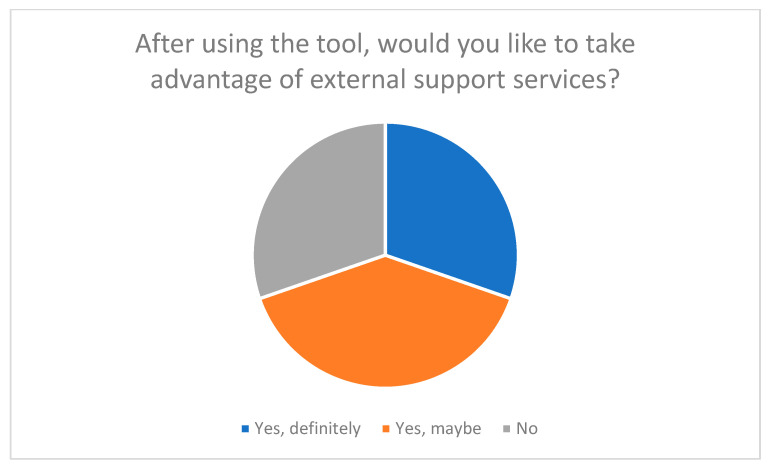
Desire to use external support services after using the tool.

**Figure 6 nursrep-15-00385-f006:**
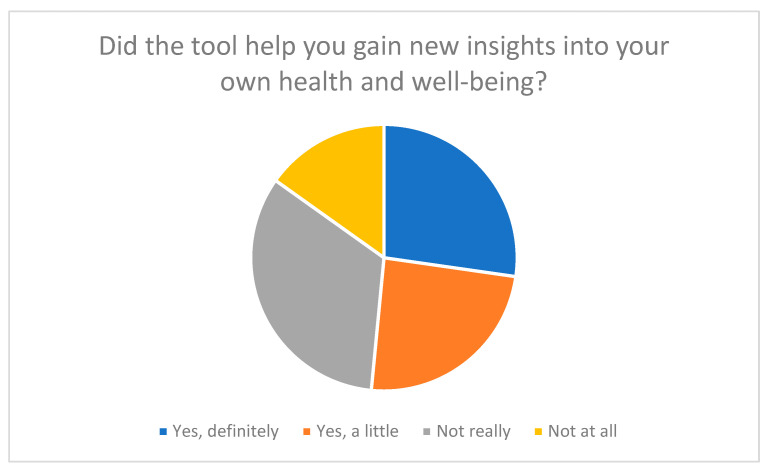
New insights into health and well-being gained through the tool.

**Figure 7 nursrep-15-00385-f007:**
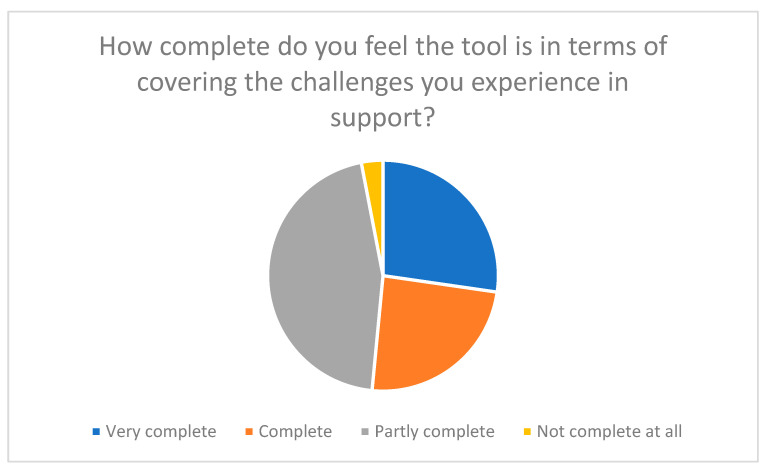
Assessment of the completeness of the tool with regard to the challenges experienced in support work.

**Figure 8 nursrep-15-00385-f008:**
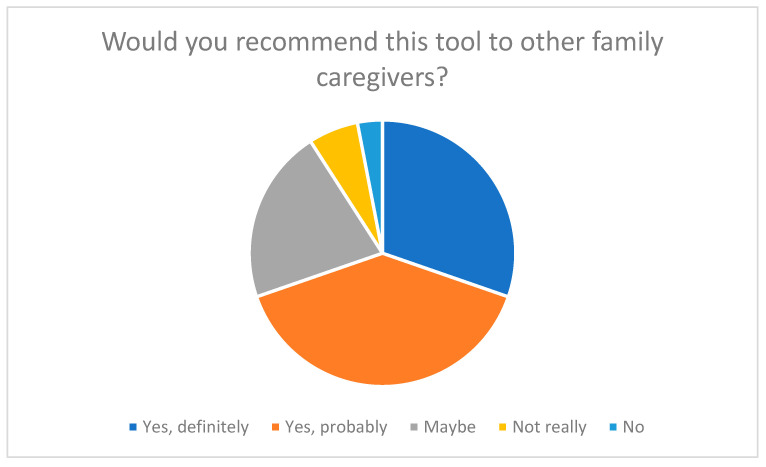
Recommendation of the tool to other family caregivers.

**Table 1 nursrep-15-00385-t001:** Overview of the SSA-PA questions for family caregivers, broken down by dimension, question, and response scale (Likert scale, multiple choice, or open-ended questions).

Dimension	Question	Response Scale
health status and well-being	1. How often do you feel physically exhausted?	Likert scale (Very rarely–Very often)
	2. How would you rate your general state of health?	Likert scale (Very good–Very bad)
	3. How often do you experience emotional exhaustion (e.g., anxiety, sadness)?	Likert scale (Very rarely–Very often)
	4. Do you regularly have time for physical activities (e.g., sports, walks)?	Likert scale (Always–Never)
	5. How often do you experience pain that interferes with your caregiving work?	Likert scale (Very rarely–Very often)
	6. Do you feel that you get enough sleep?	Likert scale (Always–Never)
	7. How often do you have time for your own health needs?	Likert scale (Very rarely–Very often)
	8. Do you feel that your diet is healthy and balanced?	Likert scale (Always–Never)
burden and stress	9. How often do you feel stressed by your caregiving responsibilities?	Likert scale (Very rarely–Very often
	10. How much do your caregiving responsibilities interfere with your sleep?	Likert scale (Not at all–Very strongly)
	11. How often do you feel that your caregiving responsibilities limit your social activities?	Likert scale (Very rarely–Very often
	12. How well are you able to cope with the stress caused by caregiving?	Likert scale (Very good–Very bad)
	13. How often do you feel that your caregiving responsibilities overwhelm you?	Likert scale (Very rarely–Very often
	14. How often do you think about giving up your caregiving responsibilities?	Likert scale (Very rarely–Very often
	15. How much emotional support do you feel you receive from friends or family?	Likert scale (Not at all–Very strongly)
	16. How often do you feel that your own health suffers as a result of your caregiving responsibilities?	Likert scale (Very rarely–Very often
support needs and resources	17. Which of the following professional care services do you use?	Multiple choice (5 possible answers)
	18. How often do you receive support from family or friends?	Multiple choice (5 possible answers)
	19. How easy is it for you to find information about care and support?	Likert scale (Very easy–Very difficult)
	20. Which of the following support services would you like to use?	Multiple choice (5 possible answers)
	21. Which of the following respite services do you use regularly? (Multiple selections possible)	Multiple choice (6 possible answers)
	22. How well informed do you feel about financial support options for family caregivers?	Likert scale (Very well informed–Not informed at all)
	23. What additional support services would improve your caregiving situation?	Open question (free text field)
	24. Do you have access to technical support or aids for caregiving (e.g., nursing beds, wheelchairs)?	Multiple choice (3 possible answers)
satisfaction and quality of life	25. How satisfied are you with the support services available for family caregivers?	Likert scale (Very satisfied–Very dissatisfied)
	26. To what extent do your caregiving responsibilities limit your professional development?	Likert scale (Not at all–Very much)
	27. How satisfied are you with your social life (friends, leisure activities)?	Likert scale (Very satisfied–Very dissatisfied)
	28. How often do you feel that your caregiving work is valued?	Likert scale (Very rarely–Very often)
	29. How often do you feel isolated or lonely because of your caregiving responsibilities?	Likert scale (Very rarely–Very often)
	30. How satisfied are you with your life overall?	Likert scale (Very satisfied–Very dissatisfied)
	31. How satisfied are you with the support you receive from state or local authorities?	Likert scale (Very satisfied–Very dissatisfied)
Additional free text fields	32. Please describe any particular challenges you experience in your caregiving role.	Open question (free text field)
	33. What are the biggest challenges you experience in your caregiving work?	Open question (free text field)
	34. What would you like to see from society or politicians to improve your situation?	Open question (free text field)
	35. What positive experiences have you had in your role as a family caregiver?	Open question (free text field)
	36. What kind of support or relief would be most important to you?	Open question (free text field)
	37. Are there specific topics or areas about which you need more information or training?	Open question (free text field)

**Table 2 nursrep-15-00385-t002:** Item-Level CVI (I-CVI) and Overall Scale Results.

Item	I-CVI
1	1
2	1
3	0.666666667
4	0.666666667
5	0.666666667
6	1
7	1
8	1
9	1
10	0.666666667
11	1
12	1
13	1
14	0.333333333
15	1
16	1
17	0.666666667
18	1
19	1
20	1
21	0.666666667
22	1
23	1
24	1
25	1
26	0.666666667
27	1
28	1
29	0.333333333
30	1
31	0.666666667
32	1
33	1
34	1
35	1
36	1
37	1

**Table 3 nursrep-15-00385-t003:** Demographic characteristics and caregiving experiences of survey participants.

Question	Response Option	Frequency	Percentage
Age of respondents	30–39	10	30.3
	40–49	9	27.3
	Under 30	6	18.2
	50–59	6	18.2
	60–69	1	3
	70	1	3
Gender	Female	26	78.8
	Male	6	18.2
	Various	1	3
Highest level of education	Vocational training	12	36.4
	University degree	7	21.2
	High school diploma	5	15.2
	Secondary school leaving certificate	4	12.1
	Other	3	9.1
	Secondary school diploma	2	6.1
Length of time caring for a relative in need of care	Less than 6 months	11	33.3
	More than 5 years	10	30.3
	1–5 years	7	21.2
	6–12 months	5	15.2
Care required per week	10–20 h	16	48.5
	More than 30 h	7	21.2
	Less than 10 h	5	15.2
	21–30 h	5	15.2

**Table 4 nursrep-15-00385-t004:** Summary of feedback on user-friendliness and comprehensibility.

Category	Positive Feedback	Suggestions for Improvement
User-friendliness	Easy to use, clear layout, clear structure of questions	Reduction in the total number of questions (items), reduction in processing time, clearer instructions on the purpose of the tool
Comprehensibility	Questions are mostly easy to understand and formulated in everyday language	Standardization of some terms, more precise explanation of the scale (e.g., rating scale 1–10)
Visual presentation	Structured presentation of the topic blocks	Addition of a progress bar, use of color accents for better orientation
Technical implementation	Stable use without major problems	Option for multiple selections for matching questions, clearer navigation at the end of the tool
Time required	Reasonable processing time for experienced users	Reduction in the total duration for first-time users, option for temporary storage if necessary
Feedback/support	Informative and self-explanatory	Provision of regional contact addresses or links to further information after completion

**Table 5 nursrep-15-00385-t005:** Relevance of the questions, including suggestions for improvement.

Category	Positive Feedback	Suggestions for Improvement
Relevance of content	Most questions perceived as appropriate and meaningful	Adaptation of some questions to different care situations (e.g., intensity of care, working caregivers)
Scope of questions	Broad coverage of topics is appreciated	Streamlining of the questionnaire to avoid repetition
Answer formats	Generally clear and understandable	Introduction of multiple-choice options if several answers apply
Individualization	Questions encourage personal reflection	Customizable topic blocks depending on the care situation or interests
Transparency of benefits	Questions perceived as meaningful	Clearer communication about how the results are used and what added value they offer

**Table 6 nursrep-15-00385-t006:** Qualitative feedback to gain new insights into one’s own health.

Category	Positive Feedback	Suggestions for Improvement
Self-reflection & mindfulness	The tool encourages self-observation and reflection on one’s own stress levels	Greater emphasis on self-care in the evaluations
Becoming aware of one’s own situation	Users report greater awareness of their role and limits	Addition of individual feedback after completion (e.g., reference to support services)
Health awareness	Some gained new perspectives on personal health	More specific recommendations for promoting health and well-being
Motivation to change	Some participants felt motivated to take better care of themselves	Integration of examples or tips for self-care after completion
Neutral/no change	For some, this was not a new insight, but still an informative experience	Clearer communication of the purpose and added value of the tool

**Table 7 nursrep-15-00385-t007:** Suggestions for expanding individual topics.

Category	Positive Feedback	Suggestions for Improvement
Social and legal issues	Questions about social networks were found to be useful	Addition of legal and social aspects (e.g., nursing care law, financial support)
Care-specific aspects	Partial consideration given	In-depth coverage of nursing topics, e.g., dealing with stress, nursing care organization
Social dimensions	Holistic approach welcomed	Expansion to include spiritual, ecological, and political aspects
Regional support	Reference to support services perceived positively	Integration of references to regional help and counseling centers
Technological issues	Questions about technical support helpful	Addition of digital tools, assistance systems, or apps for caregivers

## Data Availability

The data are contained within the article.

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
