# Peer review of "Development and Evaluation of a Self-Assessment Tool for Family Caregivers: A Step Toward Empowering Family Members"

_nursrep, 2025, doi:10.3390/nursrep15110385_

Round 1
Reviewer 1 Report
Comments and Suggestions for Authors
Dear authors,
This article addresses a very important and often overlooked topic: family caregivers.
In the introduction, I would include all the aspects that indicate the need to develop and validate a questionnaire, but I would not explain anything about the dimensions of the questionnaire or the methodology.
I would define the objective more clearly.
Although this is not a systematic review, given the importance of the literature search, the methodology should indicate the type of search conducted: sources used, search terms, etc.
Who was involved in writing the questions besides the nine family caregivers?
It should indicate who the experts were who analyzed the content validity, and what subject matter they excelled in. It should also indicate the overall Content Validity Index (CVI) for each question, and for the questionnaire in general, to assess the degree of agreement among them on the relevance of the questionnaire questions.
The different response options for Likert-type and multiple-choice questions should be indicated. How many response options do these multiple-choice questions have?
I think a table should be created with all the questions, indicating whether they are Likert-type, multiple-choice, or open-ended to facilitate the reader's understanding. It should also include which dimension each question corresponds to.
Cronbach's alpha test is used to analyze the reliability of the questionnaire, but not its validity, although only in part because it would also be necessary to assess homogeneity and test-retest reliability. What sample was used for this calculation? Was the reliability of the open-ended questions calculated? Was this test performed by dimension?
Furthermore, such a high Cronbach's alpha result, above 0.95, may indicate duplication of some items and the need to review them to assess the need to eliminate one of the duplicates. Have you taken this into account?
The previous test is not useful for analyzing the construct validity of the questionnaire; it would have been necessary to perform an exploratory or confirmatory factor analysis with Likert-type responses and perhaps with multiple-choice responses. However, how do you assess the validity of the open-ended questions?
What type of sampling was used to obtain the sample used to evaluate the questionnaire?
Why did they evaluate 8 Likert-scale questions, 5 multiple-choice questions, and 13 open-ended questions? I don't understand this evaluation, as if the questionnaire consisted of 25 Likert-scale questions, 5 multiple-choice questions, and 7 open-ended questions.
It should be indicated what statistical tests were used to analyze the Likert-scale or multiple-choice questions, and how the data obtained from the open-ended questions were analyzed.
Why are there paragraphs about the theoretical framework in the results? I believe this section is exclusively for presenting the results obtained.
How was it determined that the questionnaire was composed of four dimensions? Were they derived from theory and then developed? How was it verified that the questionnaire consisted of those four dimensions?
What does it mean that the questions were designed with validity and reliability in mind?
The results include a descriptive statistical analysis of the questions, but it does not indicate how the existence of the four dimensions was verified. Nor does it indicate the categories derived from the qualitative analysis of the open-ended questions.
The discussion is limited to analyzing the results obtained, although with few comparisons with previous studies.
The conclusions are appropriate for the results obtained, but they seem to be qualitative research because they do not address the reliability and validity of the questionnaire.
Frankly, I don't understand if what they want is: To analyze all the psychometric properties of the questionnaire? To analyze only its content validity and ease of use? To conduct a qualitative analysis of the questionnaire? To analyze the Likert-type and multiple-choice questions quantitatively, and the open-ended questions qualitatively? From my point of view, the qualitative aspect is adequate, although the quantitative aspect lacks depth since the analysis is not performed as previously indicated in the comments on the methodology.
Kind regards.
Author Response
Comments 1: This article addresses a very important and often overlooked topic: family caregivers.
Response 1: Thank you very much for your comment and positive feedback on the topic.
Comments 2: In the introduction, I would include all the aspects that indicate the need to develop and validate a questionnaire, but I would not explain anything about the dimensions of the questionnaire or the methodology.
Response 2: Thanks for the comment. The introduction has been completely revised, and the explanations of the dimensions have been removed and explained in more detail in the methodology section.
Comments 3: I would define the objective more clearly.
Response 3: Thank you for your comment. The objective has been defined more clearly.
Comments 4: Although this is not a systematic review, given the importance of the literature search, the methodology should indicate the type of search conducted: sources used, search terms, etc.
Response 4: Thank you for your comment. The methodology has been revised with regard to the literature review.
Comments 5: Who was involved in writing the questions besides the nine family caregivers?
Response 5: Thank you for your comment. It is now clearer how the tool was developed.
Comments 6: It should indicate who the experts were who analyzed the content validity, and what subject matter they excelled in. It should also indicate the overall Content Validity Index (CVI) for each question, and for the questionnaire in general, to assess the degree of agreement among them on the relevance of the questionnaire questions.
Response 6: Thank you for your comment. The experts have now been mentioned more clearly and the CVI has been specified.
Comments 7: The different response options for Likert-type and multiple-choice questions should be indicated. How many response options do these multiple-choice questions have? I think a table should be created with all the questions, indicating whether they are Likert-type, multiple-choice, or open-ended to facilitate the reader's understanding. It should also include which dimension each question corresponds to.
Response 7: Thank you for your comment. A table has been added showing all questions, including the answer scale and the respective answer options.
Comments 8: Cronbach's alpha test is used to analyze the reliability of the questionnaire, but not its validity, although only in part because it would also be necessary to assess homogeneity and test-retest reliability. What sample was used for this calculation? Was the reliability of the open-ended questions calculated? Was this test performed by dimension? Furthermore, such a high Cronbach's alpha result, above 0.95, may indicate duplication of some items and the need to review them to assess the need to eliminate one of the duplicates. Have you taken this into account?
Response 8: Thank you for your comment. The type of validation and evaluations that were carried out has been defined more explicitly. Cronbach's alpha has also been discussed in greater detail.
Comments 9: The previous test is not useful for analyzing the construct validity of the questionnaire; it would have been necessary to perform an exploratory or confirmatory factor analysis with Likert-type responses and perhaps with multiple-choice responses. However, how do you assess the validity of the open-ended questions?
Response 9: Thank you very much for your comment. This comment was also taken up and a detailed explanation of the procedure was provided in the article.
Comments 10: What type of sampling was used to obtain the sample used to evaluate the questionnaire?
Response 10: Thank you very much for your comment. This comment was also taken up and a detailed explanation of the procedure was provided in the article.
Comments 11: Why did they evaluate 8 Likert-scale questions, 5 multiple-choice questions, and 13 open-ended questions? I don't understand this evaluation, as if the questionnaire consisted of 25 Likert-scale questions, 5 multiple-choice questions, and 7 open-ended questions.
Response 11: Thank you for your comment. The presentation of the individual response scales is now clearer.
Comments 12: It should be indicated what statistical tests were used to analyze the Likert-scale or multiple-choice questions, and how the data obtained from the open-ended questions were analyzed.
Response 12: Thank you for your comment. This is explained in more detail in the methodology section.
Comments 13: Why are there paragraphs about the theoretical framework in the results? I believe this section is exclusively for presenting the results obtained.
Response 13: Thank you for your comment. The results section has been completely revised so that only the results are presented.
Comments 14: How was it determined that the questionnaire was composed of four dimensions? Were they derived from theory and then developed? How was it verified that the questionnaire consisted of those four dimensions?
Response 14: Thank you for your comment. This was derived from the theory used and feedback from experts, and is explained in more detail in the methodology section.
Comments 15: What does it mean that the questions were designed with validity and reliability in mind?
Response 15: Thank you for your comment. This has been clarified in the manuscript
Comments 16: The results include a descriptive statistical analysis of the questions, but it does not indicate how the existence of the four dimensions was verified. Nor does it indicate the categories derived from the qualitative analysis of the open-ended questions.
Response 16: Thank you for your comment. This was derived from the theory used and feedback from experts, and is explained in more detail in the methodology section.
Comments 17: The discussion is limited to analyzing the results obtained, although with few comparisons with previous studies.
Response 17: Thank you for your comment. The discussion has also been completely revised and a comparison has been included.
Comments 18: The conclusions are appropriate for the results obtained, but they seem to be qualitative research because they do not address the reliability and validity of the questionnaire.
Response 18: Thank you for your comment. The conclusion has also been revised.
Comments 19: Frankly, I don't understand if what they want is: To analyze all the psychometric properties of the questionnaire? To analyze only its content validity and ease of use? To conduct a qualitative analysis of the questionnaire? To analyze the Likert-type and multiple-choice questions quantitatively, and the open-ended questions qualitatively? From my point of view, the qualitative aspect is adequate, although the quantitative aspect lacks depth since the analysis is not performed as previously indicated in the comments on the methodology.
Response 19: Thank you very much for your comment. The questions you raised are valid and have helped us to revise the manuscript. We have addressed the issues you raised and revised the manuscript so that it is now easier to understand.
Reviewer 2 Report
Comments and Suggestions for Authors
Abstract: Include the significance/problem you are trying to address. Why is this important? Why now? Is this tool author developed or previously developed with validity and reliability? Cronbach alpha is should be rounded to the 0.998. The background and conclusion should support one another.
Introduction: Following the title, a project opens with an introduction that presents an overview of the problem, the nature, and significance of the problem, and available knowledge. The last line of the introduction should provide a description of the purpose of the project, including specific aims. There is a discussion on the tool which belongs in the methods section. The introduction should talk about what the reader is about to read (other headings). The introduction is far too dense and should be broken into more paragraphs. Although, I believe this entire introduction needs to be redone. It is not a true introduction. You state your aim is - . This tool was developed with the aim of providing family caregivers with a comprehensive and structured method for systematically recording and analyzing their individual needs, burdens, and available resources – but your title does not reflect this. I would recommend revising title to reflect your paper.
Materials and Methods: Developing a validated tool takes far more rigor than what you have described. It requires iterative changes, content reviewers, pilot testing, psychometric validation. You begin by stating a lit review was completed, but you do not provide any details on HOW the search took place, lit was selected or evaluated. You need to provide a systematic search strategy, including databases used, inclusion/exclusion criteria, how the studies were appraised. Essentially, the methodology is missing here.
Author Response
Comments 1: Abstract: Include the significance/problem you are trying to address. Why is this important? Why now? Is this tool author developed or previously developed with validity and reliability? Cronbach alpha is should be rounded to the 0.998. The background and conclusion should support one another.
Response 1: Thank you for your comment. The summary has been completely revised. Its significance and importance have been addressed. It has also been clearly stated that the tool was developed by the author. Cronbach's alpha has also been rounded.
Comments 2: Introduction: Following the title, a project opens with an introduction that presents an overview of the problem, the nature, and significance of the problem, and available knowledge. The last line of the introduction should provide a description of the purpose of the project, including specific aims. There is a discussion on the tool which belongs in the methods section. The introduction should talk about what the reader is about to read (other headings). The introduction is far too dense and should be broken into more paragraphs. Although, I believe this entire introduction needs to be redone. It is not a true introduction. You state your aim is - . This tool was developed with the aim of providing family caregivers with a comprehensive and structured method for systematically recording and analyzing their individual needs, burdens, and available resources – but your title does not reflect this. I would recommend revising title to reflect your paper.
Response 2: Thank you for your comment. The introduction has also been completely revised. The explanation of the tool has been included in the methodology section. Paragraphs have also been added, and the objective and work have been presented in such a way that the title is now more appropriate.
Comments 3: Materials and Methods: Developing a validated tool takes far more rigor than what you have described. It requires iterative changes, content reviewers, pilot testing, psychometric validation. You begin by stating a lit review was completed, but you do not provide any details on HOW the search took place, lit was selected or evaluated. You need to provide a systematic search strategy, including databases used, inclusion/exclusion criteria, how the studies were appraised. Essentially, the methodology is missing here.
Response 3: Thank you for your comment. The methodology has been fundamentally revised and all relevant content relating to literature research, validation, data analysis, and development of the tool has been addressed.
Reviewer 3 Report
Comments and Suggestions for Authors
Thank you for the opportunity to review the article “Development and evaluation of a self-assessment tool for family caregivers: A step toward empowering family members.”
The article addresses an important and timely topic, namely the need for systematic support for family caregivers—people who care for their loved ones on a daily basis, often at the expense of their own health, well-being, and professional life. The authors rightly point out that these caregivers play a key role in the long-term care and social care system, and their situation should be monitored and assessed using appropriate tools. One of the main strengths of the article is its emphasis on the importance of self-assessment tools as a way of increasing caregivers' awareness of their own needs, limitations, and resources. The concept of developing a structured tool for recording and analyzing their functional status is a step in the right direction—it can help both caregivers themselves and support institutions in adapting forms of assistance.
Despite its valuable approach, in the reviewer's opinion, the article needs to be clarified in several important aspects:
1. Study group – only family caregivers or all caregivers of dependent persons, including informal caregivers?
2. The article does not sufficiently elaborate on the potential challenges and limitations associated with the practical use of the tool. It is unclear whether caregivers are able to complete it on their own (e.g., with cognitive or emotional limitations), what the barriers to implementation are (technical, educational, time-related), and what the issue of cultural and linguistic adaptation looks like in different environments.
3. One of the most serious shortcomings of the publication is the lack of a clear reference to a specific research tool that would be analyzed or presented by the authors. The text uses vague terms – it refers to a “self-assessment tool,” but it is unclear whether it is a completely new tool or an adaptation of an existing one. There is no information about its name or structure. Meanwhile, a precise indication of the analyzed tool is crucial from the point of view of transparency and scientific value of the work.
4. The description of the tool is fragmentary and superficial. It does not specify how many items it contains, what its main areas of assessment are (e.g., physical health, emotional stress, time burden), how to interpret the results, and what they mean for further support from the caregiver.
5. The lack of numerical data significantly reduces the quality of the presentation of the results – this should be supplemented in the figures.
Summary: The article addresses an extremely important topic and makes a valuable contribution to the discussion on the needs of family caregivers/informal caregivers. The concept of creating a tool for self-assessment of caregivers' functional status is valid and necessary. However, from a scientific point of view, the publication needs to be supplemented with empirical data and a more detailed methodological description. Only then will the tool be able to be evaluated in terms of its usefulness, reliability, and potential impact on the care system.
Author Response
Comments 1: Thank you for the opportunity to review the article “Development and evaluation of a self-assessment tool for family caregivers: A step toward empowering family members.”
The article addresses an important and timely topic, namely the need for systematic support for family caregivers—people who care for their loved ones on a daily basis, often at the expense of their own health, well-being, and professional life. The authors rightly point out that these caregivers play a key role in the long-term care and social care system, and their situation should be monitored and assessed using appropriate tools. One of the main strengths of the article is its emphasis on the importance of self-assessment tools as a way of increasing caregivers' awareness of their own needs, limitations, and resources. The concept of developing a structured tool for recording and analyzing their functional status is a step in the right direction—it can help both caregivers themselves and support institutions in adapting forms of assistance.
Despite its valuable approach, in the reviewer's opinion, the article needs to be clarified in several important aspects:
Response 1: Thank you very much for your comment and positive feedback on this important topic.
Comments 2: 1. Study group – only family caregivers or all caregivers of dependent persons, including informal caregivers?
Response 2: Thank you for your comment. The study group has been clarified. It refers to all family caregivers.
Comments 3: 2. The article does not sufficiently elaborate on the potential challenges and limitations associated with the practical use of the tool. It is unclear whether caregivers are able to complete it on their own (e.g., with cognitive or emotional limitations), what the barriers to implementation are (technical, educational, time-related), and what the issue of cultural and linguistic adaptation looks like in different environments.
Response 3: Thank you for your comment. The challenges have been outlined, so it is now clearer what obstacles may exist when using the tool.
Comments 4: 3. One of the most serious shortcomings of the publication is the lack of a clear reference to a specific research tool that would be analyzed or presented by the authors. The text uses vague terms – it refers to a “self-assessment tool,” but it is unclear whether it is a completely new tool or an adaptation of an existing one. There is no information about its name or structure. Meanwhile, a precise indication of the analyzed tool is crucial from the point of view of transparency and scientific value of the work.
Response 4: Thank you for your comment. It is now clearer that this is a new tool that has been developed and evaluated. In addition, the name of the tool has been integrated and the content and individual items have been presented more precisely.
Comments 5: 4. The description of the tool is fragmentary and superficial. It does not specify how many items it contains, what its main areas of assessment are (e.g., physical health, emotional stress, time burden), how to interpret the results, and what they mean for further support from the caregiver.
Response 5: Thank you for your comment. All content has been revised and feedback from all reviewers has been taken into account. Among other things, this means that it is now clearer how the items were developed and which items are available.
Comments 6: 5. The lack of numerical data significantly reduces the quality of the presentation of the results – this should be supplemented in the figures.
Response 6: Thank you for your comment. The presentation of the results has been revised.
Comments 7: Summary: The article addresses an extremely important topic and makes a valuable contribution to the discussion on the needs of family caregivers/informal caregivers. The concept of creating a tool for self-assessment of caregivers' functional status is valid and necessary. However, from a scientific point of view, the publication needs to be supplemented with empirical data and a more detailed methodological description. Only then will the tool be able to be evaluated in terms of its usefulness, reliability, and potential impact on the care system.
Response 7: Thank you very much for your helpful comments. The work has been fundamentally revised, taking into account all comments from all reviewers.
Reviewer 4 Report
Comments and Suggestions for Authors
Keywords
The terms used are neither mesh nor decs terms, and the authors should update them.
Introduction
Part of the introduction should be devoted to presenting existing knowledge about the tools currently available for measuring caregiver and family fatigue, since the objective is to create a new tool for this type of population.
Objective
The objective should be clearly stated in one sentence.
Method
The academic background of the experts, their job position, years of experience, age, and gender should be stated.
The time frame of the study, including its start and end dates, should also be identified.
Reference is made to the inclusion of free text fields; these should be identified.
The characteristics of the literature review conducted to determine the measurement dimensions of the tool developed should be presented.
The sociodemographic characteristics of the family group that evaluated the tool and were part of the caregiving course should also be described.
Results
In section 3.1, the first and second paragraphs form part of the method, not the results.
Although pie charts are used, it is important to include the response number for each section of the chart in order to better appreciate the results.
Table 3 corresponds to the section on method, not results.
The sentence “Overall, the survey confirms that the self-assessment tool is a valuable and well-received instrument for family caregivers” does not correspond to the results in Figures 1, 2, 3, and 4, which say the opposite.
Discussion
The statement “The survey confirmed the user-friendliness and clarity of the tool's questions, underscoring its intuitive use and usefulness for users” does not correspond to the results in Figures 1, 2, 3, and 4, which indicate the opposite.
There should be a section reflecting on the results of Figures 1, 2, 3, and 4, which indicate that the usability and clarity of the tool are low and its difficulty is high. These are aspects that should not occur in the development of a new tool, because otherwise it cannot be validated as it is not valid for clinical use.
Conclusion
The statement “The evaluation has shown that the tool is useful and relevant and offers valuable insights for further development” does not correspond to the results of Figures 1, 2, 3, and 4.
Author Response
Comments 1: Keywords
The terms used are neither mesh nor decs terms, and the authors should update them.
Response 1: Thank you very much for your helpful comments. The terms have been revised.
Comments 2: Introduction
Part of the introduction should be devoted to presenting existing knowledge about the tools currently available for measuring caregiver and family fatigue, since the objective is to create a new tool for this type of population.
Response 2: Thank you very much for your helpful comments. The introduction has been revised and explicitly addresses the available instruments as well as the need to develop further tools.
Comments 3: Objective
The objective should be clearly stated in one sentence.
Response 3: Thank you for your comment. The goal has been clearly stated.
Comments 4: Method
The academic background of the experts, their job position, years of experience, age, and gender should be stated.
The time frame of the study, including its start and end dates, should also be identified.
Reference is made to the inclusion of free text fields; these should be identified.
The characteristics of the literature review conducted to determine the measurement dimensions of the tool developed should be presented.
The sociodemographic characteristics of the family group that evaluated the tool and were part of the caregiving course should also be described.
Response 4: Thank you for your comment. The methodology has been fundamentally revised. The academic background has been addressed and the time frame has been supplemented. The individual items, including the free text fields, have also been presented more precisely. The literature review has been explained in more detail and the demographic data of the participants has been presented.
Comments 5: Results
In section 3.1, the first and second paragraphs form part of the method, not the results.
Although pie charts are used, it is important to include the response number for each section of the chart in order to better appreciate the results.
Table 3 corresponds to the section on method, not results.
The sentence “Overall, the survey confirms that the self-assessment tool is a valuable and well-received instrument for family caregivers” does not correspond to the results in Figures 1, 2, 3, and 4, which say the opposite.
Response 5: Thank you for your comment. The results have also been fundamentally revised so that only aspects of the presentation of results are integrated. The results of the responses have also been presented more clearly, as has the interpretation of the data. Suggestions for improvement were discussed and incorporated into the evaluation.
Comments 6: Discussion
The statement “The survey confirmed the user-friendliness and clarity of the tool's questions, underscoring its intuitive use and usefulness for users” does not correspond to the results in Figures 1, 2, 3, and 4, which indicate the opposite.
There should be a section reflecting on the results of Figures 1, 2, 3, and 4, which indicate that the usability and clarity of the tool are low and its difficulty is high. These are aspects that should not occur in the development of a new tool, because otherwise it cannot be validated as it is not valid for clinical use.
Response 6: Thank you very much for your comment. The results of the responses were presented more clearly, as was the interpretation of the data. Suggestions for improvement were discussed and incorporated into the evaluation.
Comments 7: Conclusion
The statement “The evaluation has shown that the tool is useful and relevant and offers valuable insights for further development” does not correspond to the results of Figures 1, 2, 3, and 4.
Response 7: Thank you very much for your comment. The results of the responses were presented more clearly, as was the interpretation of the data. Suggestions for improvement were discussed and incorporated into the evaluation.
Round 2
Reviewer 1 Report
Comments and Suggestions for Authors
Dear authors,
I consider the manuscript sufficiently improved and have no further comments or suggestions.
Kind regards.
Author Response
Comments 1: Dear authors,
I consider the manuscript sufficiently improved and have no further comments or suggestions.
Kind regards.
Response 1: Thank you very much for your kind feedback and for acknowledging the improvements made to the manuscript.
Reviewer 4 Report
Comments and Suggestions for Authors
The revisions made by the authors are adequate and have responded to the requested clarifications. However, it is necessary to review the following aspects of the methodology:
The chronological sequence of each of the three phases carried out in the development of the study design must be indicated: (1) theoretical conceptualization and literature review, (2) item formulation and validation, and (3) evaluation by family caregivers.
In phase 1 of the literature review, the inclusion and exclusion criteria used should be established in a more comprehensive manner. The time interval for the search, language of the articles, and type of articles should be specified, including meta-analyses.
Author Response
Comments 1:
The revisions made by the authors are adequate and have responded to the requested clarifications. However, it is necessary to review the following aspects of the methodology:
The chronological sequence of each of the three phases carried out in the development of the study design must be indicated: (1) theoretical conceptualization and literature review, (2) item formulation and validation, and (3) evaluation by family caregivers.
In phase 1 of the literature review, the inclusion and exclusion criteria used should be established in a more comprehensive manner. The time interval for the search, language of the articles, and type of articles should be specified, including meta-analyses.
Response 1: Thank you very much for your valuable comments and constructive feedback on the article. The requested changes have been implemented and the methodology revised accordingly. The chronological sequence of the three phases of study development is now clearly presented: (1) theoretical conceptualization and literature review, (2) item formulation and validation, and (3) evaluation by family caregivers. In phase 1 of the literature review, the inclusion and exclusion criteria have been described in greater detail. The search period, the language of the articles, and the type of articles, including meta-analyses, have been specified.